# Valorization of Peach Fruit and Wine Lees through the Production of a Functional Peach and Grape Juice

**DOI:** 10.3390/foods13071095

**Published:** 2024-04-02

**Authors:** Virginia Prieto-Santiago, Ingrid Aguiló-Aguayo, Francisca Isabel Bravo, Miquel Mulero, Maribel Abadias

**Affiliations:** 1Institute of Agrifood Research and Technology (IRTA), Postharvest Program Edifici Fruitcentre, Parc Agrobiotech Lleida, Parc de Gardeny, 25003 Lleida, Spain; virginia.prieto@irta.cat (V.P.-S.); ingrid.aguilo@irta.cat (I.A.-A.); 2Nutrigenomics Research Group, Departament de Bioquímica i Biotecnologia, Universitat Rovira i Virgili, Marcel·lí Domingo 1, 43007 Tarragona, Spain; franciscaisabel.bravo@urv.cat (F.I.B.); miquel.mulero@urv.cat (M.M.); 3Nutrigenomics Research Group, Institut d’Investigació Sanitària Pere Virgili, C/Marcel·lí Domingo s/n, 43007 Tarragona, Spain; 4Center of Environmental, Food and Toxicological Technology (TecnATox), Universitat Rovira i Virgili, 43007 Tarragona, Spain

**Keywords:** peach, wine lees, functional, antimicrobial activity, antioxidant capacity, valorization

## Abstract

The valorization of agri-food products not only represents important economic and environmental benefits but can also be a source of potentially profitable, functional, and safe ingredients. This study aimed to valorize peach fruit and wine lees (WL) by producing functional juice. WL were incorporated at different concentrations (1.5 and 2%; *w*:*w*) in unpasteurized peach and grape juice and subsequently stored under refrigeration (5 °C). The antimicrobial activity of WL in peach and grape juices was assessed against *Listeria monocytogenes* and *Saccharomyces cerevisiae* as well as physicochemical, nutritional microbiological, and sensory acceptability. The maximum addition of WL to the juice (2%) showed a significant inhibitory effect against *L. monocytogenes* (4-log reduction) and increased the content of total soluble solids (TSS) (10%), total polyphenol content (TPC) (75%), and total antioxidant activity (AOX) (86%). During storage, AOX, TPC, TSS, pH, and titratable acidity (TA) remained stable. A significant correlation was observed between TPC and AOX. Total mesophilic aerobic bacteria and yeast counts increased during storage. Fifty-seven percent of tasters (n = 26) rated the functional juice positively. Thus, these agri-food products could be useful for producing functional juices with a longer shelf life, contributing to their valorization.

## 1. Introduction

The COVID-19 health crisis and growing concern for better self- and planet health have enhanced the public interest in food products with natural ingredients, popularizing plant-based diets, organic alternatives, and functional foods. Regarding consumer inclinations towards healthier options, some epidemiological studies have shown that the daily intake of plant-based foods could be positive for health status and could contribute to the prevention of some types of cancer of the gastrointestinal tract, cardiovascular diseases, and the risk of diabetes [1]. Functional foods not only stand out for their nutritional value but also for playing an important role in promoting optimal health, even preventing some non-communicable diseases such as cancer or cardiovascular pathologies [2]. Organic micronutrients such as carotenoids, polyphenolics, tocopherols, and others are generally responsible for these health benefits and are, mostly, attributed to fruits and vegetables [3]. Among the bioactive compounds found in fruits, some polyphenols, and vitamins potentially exhibit antioxidant, anti-inflammatory, antitumor, and antimicrobial activities [4,5]. In addition to the healthy tendency in consumption, there is a current trend in the food and beverage industry to reduce waste based on the valorization of by-products, and co-products, that contribute to a circular economy and sustainability in production [6].

Regarding the new healthy trend and the subsequent inclination towards plant-based diets and food products with functional properties, fruit is known to be rich in bioactive compounds, which have a wide range of action mechanisms, including effects on antioxidant activity and free radicals, cell cycle, oncogene expression, metabolism, infections, among others [7]. The major sources of valuable bioactive substances in fruit juices are phenolic compounds, mostly flavonoids and phenolic acids [8,9,10]. (Poly)phenols, recognized as one of the most abundant groups of bioactive compounds, are plant secondary metabolites with protective functions in plants and are present in significant quantities of vegetables and fruits [11]. However, most fruits and, in particular, peaches have a short postharvest life due to rapid ripening and their vulnerability to fungal infection. Although these challenges can be managed by refrigeration, chemical preservatives, and packaging conditions, consumer acceptance is limited, leading to massive fruit losses. Therefore, also to avoid wastage in years of overproduction or because of aesthetic defects, juices can be a good alternative to whole fruit consumption, valorizing fruits that are not suitable for fresh consumption but have the same nutritional qualities.

Nutritionally, juices are also an alternative to fruit consumption because they are also rich in vitamins and minerals, which contribute to human health. Vitamins contained in fruit juices with antioxidant properties, such as ascorbic acid (vitamin C), can potentially work synergistically with the phenolic compounds, increasing the total antioxidant activity of the final product [12]. However, these bioactive substances are usually reduced during thermal treatment. In this regard, besides non-thermal technologies, fresh, minimally processed juices with high nutritional value and rich in functional and antimicrobial substances could be an option to deliver nutritious foods in response to the evolving consumer demands and a trend towards healthier foods. Clinical studies have demonstrated that the consumption of juices with bioactive properties may provide health benefits that prevent biomarkers related to lipid oxidation, decrease high blood pressure, and improve cognitive function and quality of life indices [13].

The main safety concerns in the beverage industry are related to the contamination and growth of pathogenic or spoilage microorganisms. In particular, acid-tolerant human pathogenic bacteria such as *Listeria monocytogenes*, and spoilage microorganisms such as the yeast *Saccharomyces cerevisiae* can lead to foodborne disease or undesirable fermentation, and, thus, the production of some compounds, changing the physicochemical properties of the juice and yielding rejection and safety issues [14]. Phenolic compounds and other phytochemicals, along with low pH, may influence the antimicrobial properties of juices [9]. The ability of some fruit compounds to inhibit the growth of different human pathogenic bacteria has already been reported [15,16].

The utilization of grapes in the wine industry generates by-products such as seeds, skins, and lees. Wine lees (WL) are the remnants formed at the bottom of the wine tank after fermentation. This by-product is one of the least valorized, but it is very interesting as it contains high levels of grape (poly)phenols [17,18]. Previous studies have reported the antioxidant, anti-inflammatory, and cardioprotective properties of WL [17,18,19]. Moreover, the soluble fraction of WL also exerted antihypertensive properties in spontaneously hypertensive rats, whose effects were associated with their content of anthocyanins and flavanols [20,21]. Other studies have also highlighted other functionalities of WL that are related to their antimicrobial capacity. The important concentration of (poly)phenols, particularly anthocyanins, is suggested to be the main contributor to the antimicrobial effect of red wine lees against some pathogens, such as *L. monocytogenes* [22,23]. Due to their high nutritional value and the feasible antimicrobial activity, WL could be considered a great option for their valorization within the sustainability framework [17].

This study aimed to explore the potential of peaches and to valorize WL as a functional ingredient within a circular economy to improve the nutritional, physicochemical, and microbiological quality of fresh, unpasteurized, and refrigerated peach and grape juice.

## 2. Materials and Methods

### 2.1. Materials

#### 2.1.1. Peach and Grape Juice

Peach juice was obtained from yellow-flesh peach (*Prunus persica*/L./Batch) cv. Royal Summer, in the pilot plant at IRTA’s Fruitcentre facilities, using an automatic extractor (Robot Coupe C80A, Mataró, Barcelona, Spain). This juice contained neither added sugars nor was it subjected to subsequent clarification or pasteurization treatments. The juice was kept frozen (at −20 °C) until further use.

Pasteurized grape juice (ECO-Call Valls, Vilanova de Bellpuig, Lleida, Spain), with no added sugars, was elaborated directly from organically grown grapes of the native ‘Macabeu’ and ‘Parellada’ varieties.

The defrosted peach juice and commercial grape juice were mixed 50:50 to obtain the peach and grape juice (PGJ).

#### 2.1.2. Wine Lees

Considering that the present study includes a sensorial analysis of the juices with or without the addition of wine lees, the product DULASAVU^®^ was chosen as the source of wine lees to ensure their safety. This product is obtained after filtering and drying the soluble fraction of wine lees and was manufactured and kindly supplied by Grandes Vinos y Viñedos S.A. (Cariñena, Spain). The final product, containing 47.1% of dry wine lees and 52.9% of maltodextrin, will be referred to as “WL” throughout the manuscript. Maltodextrin was added as a support for the spray drying process.

The phenolic composition of this WL-derived product was analyzed by UHPLC-ESI-QQQ-MS using 1290 UHPLC Infinity II series coupled to a QQQ/MS 6470 (Agilent Technologies, Palo Alto, CA, USA) [24]. Samples were injected at 2.5 µL into an Acquity HSST3 C18 column (150 mm × 2.1 mm i.d., 1.8 µm particle size) and an Acquity BEH C18 column (100 mm × 2.1 mm, 1.7 µm particle size) (Waters, Milford, MA, USA) at 45 °C, 0.55 mL/min or 0.4 mL/min, respectively, for the separation of non-anthocyanin and anthocyanin compounds, respectively. For non-anthocyanin separation, mobile phases were water/acetic acid (95:5, *v*:*v*, phase A) and acetonitrile (phase B) and the flow gradient was as follows: 0% B, which remained on it for 0.5 min; 0–30% B from 0.5 to18 min; 30–95% B from 18 to 21 min; 95% B from 21 to 24 min; and 100–0% B from 24 to 25 min. For anthocyanin separation, mobile phases were water/formic acid (9:1, *v*:*v*, phase A) and acetonitrile (phase B) and the gradient elution was as follows: initial conditions, 0% B during 0.5 min; 0–9% B from 0.5 to 5 min; 9–15% B from 5 to 7 min; 15–30% B from 7 to 9.5 min; 30–100% B from 9.5 to 10 min; 100% B from 10 to 12 min; and 100–0% B from 12 to 12.1 min. Electrospray ionization (ESI) operating in negative or positive mode (for non-anthocyanin and anthocyanin detection, respectively) employed a gas temperature of 200 °C, a flow rate of 14 L/min, a nebulizer gas pressure of 20 psi, a sheath gas temperature of 350 °C, a sheath gas flow of 11 L/min, and a capillary voltage of 3000 V. Quantitative determination was carried out using the multiple reaction monitoring (MRM) mode, with specific transitions detailed in Table A1 and Table A2 for each compound. Calibration curves derived from commercial standards were utilized for quantifying corresponding phenolic compounds. For other compounds, the analysis was semi-quantitative, determining the compound’s behavior and selecting the calibration curve that best matched its characteristics for concentration calculation. Table A1 and Table A2 show the content of non-anthocyanin phenolic compounds and anthocyanin phenolic compounds, respectively. The main phenolic families found in the WL were the anthocyanins (329 mg/100 g) followed by flavanols (52 mg/100 g), flavonols (50 mg/100 g), phenolic acids (15 mg/100 g), and stilbenes (7.9 mg/100 g). The major compounds found in each family, respectively, were malvidin-3-glucoside (210 mg/100 g), which represented 64% of the total anthocyanins identified (39 compounds), several procyanidin dimers (~9 mg/100 g each compound), quercetin-3-*O*-glucuronide (41 mg/100 g), gallic acid (11 mg/100 g), and resveratrol-*O*-glucoside iso2 (2 mg/100 g).

#### 2.1.3. Formulation of Functional Juices

Different concentrations of wine by-products, 0.5, 1, 1,5, and 2% (*w*:*w*), were added to the PGJ obtained as described in Section 2.1.1.

#### 2.1.4. Culture Medium and Reagents

Tryptone soy broth (TSB), tryptone soy agar (TSA), Palcam base agar, Palcam selective supplement for *Listeria*, yeast extract, plate count agar (PCA), Dichloran Rose Bengali Chloramphenicol Agar (DRBC), and peptone were obtained from Biokar Diagnostics (Allonne, France), and Dew-Engley medium was obtained from Merck (Darmstadt, Germany).

Ascorbic acid, gallic acid, 2,4,6-tris(2-pyridyl)-s-triazine (TPTZ), sodium carbonate, metaphosphoric acid, acetic acid, and 2,2′-Azino-bis (3-ethylbenzothiazoline-6-sulfonic acid) diammonium salt (ABTS) was obtained from Sigma-Aldrich (Steinheim, Germany). Methanol, acetone, chlorhidric acid (37%), sodium acetate, sodium hydroxide, sodium chloride, potassium chloride, ferric chloride hexahydrate, and Folin–Ciocalteu’s reagent were procured by Panreac (Llinars del Valles, Spain).

### 2.2. Microorganism Strains and Preparation

The microbiological effects of the addition of different concentrations of WL to peach and grape juices were assessed following Ortiz-Solá et al. [25]. The bacterial strains used in this work included a cocktail of 5 strains of *Listeria monocytogenes*: serovar 1a (CECT 4031), serovar 3a (CECT 933), serovar 4d (CECT 940), serovar 4b (CECT 4032), and serovar 1/2a, which was previously isolated in our laboratory from a fresh-cut lettuce sample [26]. The yeast *Saccharomyces cerevisiae* WDCM00058 was used as a model for spoilage microorganisms.

Briefly, a single colony of each culture of *L. monocytogenes* grown in tryptone soy agar plus 6 g/L of yeast extract (TSAYE) was inoculated in a 50 mL Erlenmeyer flask of TSB medium supplemented with 6 g/L of yeast extract, 2.5 g/L of glucose, and 2.5 g/L of K_2_HPO_4_ (TSBYE) for 20–24 h at 37 ± 1 °C. Subsequently, 10 mL of each *L. monocytogenes* strain was centrifuged at 9800× *g* for 10 min at 10 °C (Sorvall Legend XTR Centrifuge, Thermo Fischer, Waltham, MA, USA) and then resuspended in 25 mL of sterile saline solution (SS; 0.85% *w*/*v* NaCl). Equal volumes of the five *L. monocytogenes* concentrated suspensions were mixed to provide the 5-strain concentrated cocktail.

On the other hand, *S. cerevisiae* was grown in YPD broth (5 g/L of yeast extract, 10.0 g/L of peptone, and 20.0 g/L of glucose) at 25 ± 1 °C for 48 ± 4 h, centrifuged, and resuspended in 25 mL of SS.

The concentrate suspension of each microorganism was checked by plating appropriate dilutions in YPD medium for *S. cerevisiae* and Palcam agar (Palcam Agar Base with a selective supplement, Biokar Diagnostics) for *L. monocytogenes*, followed by incubation at 37 ± 1 °C and 25 ± 1 °C, respectively for 48 ± 2 h.

### 2.3. Effect of the Addition of Wine Lees on the Survival of Listeria monocytogenes and Saccharomyces cerevisiae in Artificially Inoculated on Peach and Grape Juice

WL product was added to PGJ and prepared as described above to obtain 0.5, 1.0, 1.5, and 2.0% (*w*/*v*) juices. PGJ without WL was used as the control treatment (CK). Afterwards, a volume of the concentrated suspension (*L. monocytogenes* or *S. cerevisiae*) was added to the juice to obtain an initial concentration of approximately 10^5^ CFU/mL. Juices were distributed in 10 mL screw-cap glass tubes and stored at 5 ± 1 °C. Samples were taken at 0, 3, 6, 9, and 14 days to determine the microbial population. A series of decimal dilutions were made from the juice in saline peptone (peptone 1 g/L + NaCl 8.5 g/L) and surface-plated in the corresponding medium (YPD for *S. cerevisiae*; Palcam for *L. monocytogenes*), followed by incubation at 25 °C and 37 °C, respectively, for 48 h. Samples (1 mL) were also enriched in Dew-Engley medium (9 mL) to determine the presence or absence of the microorganisms when the counts were below the detection limit. The results were expressed as log CFU/mL.

Regarding the effect of the different doses of WL in the juice against the pathogen and spoilage microorganisms, two concentrations of WL were chosen for the evaluation of the effect of the addition of WL on the physicochemical, nutritional, microbiological, and sensory quality of PGJ juice.

### 2.4. Effect of the Addition of Wine Lees on the Quality of Peach and Grape Juice

Frozen Royal Summer peach juice (2 L) was defrosted and mixed (50:50, *v*:*v*) with the commercial organic grape juice (2 L). This juice mixture was divided into three batches: (a) a control juice (CK) corresponding to PGJ (50:50, *v*:*v*), without the addition of WL product; (b) a 1.5% WL concentration, where 19.5 g of WL was added to 1300 mL of the control juice and mixed manually until completely dissolved in the juice matrix; and (c) a 2% WL concentration, where 26× *g* of WL was added to 1300 mL of the control juice and mixed manually until completely dissolved in the juice matrix. Each juice was distributed in 50 mL sterile centrifuge tubes. Three tubes (replicates) were prepared for each treatment and sampling period. The juices (control and 1.5 and 2.0% of WL concentration) were stored refrigerated throughout the trial at 5 ± 1 °C and sampled after 0, 4, 7, 11, 14, and 21 days.

Physicochemical, nutritional, and microbiological analyses of the experimental concentrations were carried out in triplicate on the sampling days. Sensory analysis was performed after 4 days of storage at 5 °C.

#### 2.4.1. Physicochemical Analysis

Analyses of pH, titratable acidity (TA), and total soluble solids (TSS) were carried out following the methodology proposed by Nicolau-Lapeña et al. [27]. The pH and TA of the samples were determined using an automatic titrator (Titralab AT1000 Series, HACH, Vésenaz, Switzerland). The TA was determined by titration with 0.1 M of NaOH, previously diluting the experimental juices by half in distilled water (1:1, *v*/*v*). TA was expressed as g of malic acid/L. TSS content was determined using a digital refractometer (Atago Co., Ltd., Tokyo, Japan) with a range of 0–45% and expressed in °Brix.

#### 2.4.2. Nutritional Analysis

Total phenolic content (TPC): The TPC was determined using the Folin–Ciocalteu method, following the procedure described in Nicolau-Lapeña et al. [27]. For the extraction of phenolic and antioxidant compounds, samples were mixed with a solution of 70% methanol (30%, *w*/*v*) and homogenized in a vortex for 20 s. The samples were immediately placed in a stirrer at 4 °C, working at 195 rpm for 20 min, and centrifuged using a Sigma-3-18 KS centrifuge (Sigma Laborzentrifugen GmbH, Osterode am Harz, Germany) at 13,500× *g* for 20 min at 4 °C. The extracts were stored at −80 °C for further analysis.

The assay was performed by adding 0.1 mL of Folin–Ciocalteu’s reagent to 0.020 mL of extract. After shaking and incubating for 5 min at room temperature in the dark, 0.075 mL of saturated sodium carbonate was added. The mixture was once again shaken and incubated for 2 h in the dark. Absorbance was read at 760 nm using a FLUOstar Omega BMG Labtech UV–Vis spectrophotometer (Thermo Fisher Scientific, MA, USA). A standard curve of gallic acid was prepared daily using the same procedure as that used for the samples. The results were expressed as mg of gallic acid equivalents (GAE)/100 mL fresh weight (f.w).

Total antioxidant capacity (AOX): The AOX was assessed in the frozen juice extracts using two methodologies: ferric reducing antioxidant power (FRAP) and scavenging activity assay (ABTS), as previously described in Nicolau-Lapeña et al. [27] and Thaipong et al. [28], respectively. The experiments were performed using the same extract employed for the phenolic content determination. The FRAP reagent was prepared with a mixture of acetate buffer 0.3 M pH 2.6,2,4,6-tris(2-pyridyl)-s-triazine (TPTZ) 40 mM in HCl and FeCl_3_·6H_2_O 20 mM in distilled water in a 10:1:1 (*v*:*v*:*v*) proportion. The determination was performed by adding 0.02 mL of the extract to 0.180 mL of the FRAP reagent and incubating at 37 °C for 20 min in the dark. Absorbance was read at 593 nm using a FLUOstar Omega spectrophotometer (BMG Labtech, Ortenberg, Germany).

ABTS radical was prepared daily by diluting a stock solution of ABTS 7 mM in water until an absorbance at 734 nm of 0.750 ± 0.50 was reached. Then, the determination was performed by adding 0.2 mL of the extract to 0.8 mL of ABTS reagent and incubating at room temperature for 30 min in the dark. Absorbance was read at 734 nm using a FLUOstar Omega spectrophotometer (BMG Labtech, Ortenberg, Germany).

The percentage of antioxidant activity versus the different concentrations of the juice extracts was plotted.

Standard curves of ascorbic acid for both methods were prepared daily using the same procedure as with the samples. The results were expressed as a mg equivalent of ascorbic (mg EAA)/100 mL of fresh weight (f.w).

Maximum mean inhibitory concentration (IC_50_): The IC_50_ value was defined as the effective concentration at which the ABTS radical was scavenged by 50% [29]. PGJ extracts were ten-fold serially diluted in distilled water. Then, 0.02 mL of the diluted samples were placed with 0.180 mL of 0.1 Mm ABTS solution in a 96-well plate. After incubation at room temperature for 40 min in the dark, absorbance at 734 nm was read using a FLUOstar Omega spectrophotometer (BMG Labtech, Ortenberg, Germany).

To calculate the IC_50_, the inhibition percentage calculated by Equation (1) was plotted against the concentration of the extract. Thus, IC_50_ corresponded to the necessary concentration of an extract to achieve a 50% inhibition (Equation (2)).
%I = [(Ab − As)/Ab] × 100(1)

This first equation corresponds to the linear regression resulting from plot concentration versus the inversion of the absorbance; where %I is the inhibition in percent, Ab and As are the absorbances of the blank and the sample, respectively.
%I = m ∗ C + n; IC_50_ = (50 − n)/m(2)

This second equation was used for calculating the concentration (mg/L) needed for reaching a 50% inhibition; m is the slope of the linear fit when representing inhibition percentage (%I) versus concentration *C* (mg/mL) and n is the intercept. The results were expressed as mg/mL.

#### 2.4.3. Microbiological Analysis

In order to assess the microbiological quality of the different PGJs and confirm the feasible antimicrobial effect of the lees, duplicate counts of three replicates of mesophilic aerobic microorganisms, molds, and yeasts were carried out, following the methodologies ISO 4833-2:2013 [30] and ISO 21527-1:2008 [31], respectively. Briefly, decimal dilutions in peptone saline and subsequent plating (0.1 mL) were performed in duplicate on PCA for total aerobic mesophilic counts and DRBC for the molds and yeasts. The plates were incubated at 30 ± 1 °C for 3 days for the aerobic mesophiles and at 25 ± 1 °C for 3–5 days for the molds and yeasts. The results were expressed as log CFU/mL.

#### 2.4.4. Sensorial Analysis

The sensory evaluation of the juices was performed after 4 days of storage. Twenty-six semi-trained panelists (19 women and 7 men), regular consumers of fruit juice and recruited at IRTA Fruitcentre in Lleida, Spain, were included as participants. Sensory evaluation was carried out in a tasting room equipped with individual booths. Samples were served in randomly coded glasses and presented to testers in a random order. A time of 60 s was used between each sample to reduce sensory fatigue. Each panelist was asked to evaluate all samples in two aspects: taste, using a 5-point hedonic scale (from 1: extremely dislike to 5: extremely like) and overall acceptability, using a 9-point hedonic scale (from 1: extremely dislike to 9: extremely like). The acceptability index was calculated as described in previous studies [32].

### 2.5. Statistical Analysis

Results are expressed as the mean ± standard deviation (SD) of 3 repetitions. All data were checked for significant differences using the analysis of variance test (ANOVA). The criterion for statistical significance was set at *p* < 0.05. When significant differences were observed, Tukey’s Honest Significant Difference (HSD) of the means was applied. Correlation analysis of antioxidant activity values vs. phenolic content was performed using the Pearson Test. All statistical analyses were carried out using JMP 13 (SAS Institute Inc., Cary, NC, USA).

## 3. Results and Discussion

### 3.1. Effect of the Addition of Wine Lees on the Survival of Listeria monocytogenes and Saccharomyces cerevisiae Artificially Inoculated on Peach and Grape Juice

The initial population of *L. monocytogenes* in all the different peach and grape juices (PGJs) was 5.5 log CFU/mL (Figure 1). *L. monocytogenes* was not able to grow in the control juice, and significantly decreased during the 14 days of storage (2.5 log reduction), achieving the final population of 3.42 log CFU/mL. WL was demonstrated to have an antimicrobial effect with higher activity at the highest concentrations tested (1.5 and 2.0%). A reduction of almost 4 log units was achieved after 14 days of treatment in the juice with the highest WL concentration (2%).

*Saccharomyces cerevisiae* was able to grow in the PGJ juice (Figure 2), with an increase of almost 2 log units after 14 days of storage. The growth of *S. cerevisiae* was slightly reduced (<1 log) in the juices containing the WL at 1.0, 1.5, and 2.0%.

Unpasteurized fruit juice safety concerns are related to acid-tolerant bacteria, such as *L. monocytogenes*, and spoilage microorganisms, such as yeast like *S. cerevisiae* [14].

The evaluation of the in vitro antimicrobial potential of WL using the microdilution method has already been studied by Tagkouli et al. [22], who demonstrated the strong antioxidant and antimicrobial activity of wine lees from winemaking by-products. The antibacterial effect that emerged from the red wine lees is attributed to the high concentration of anthocyanins and other (poly)phenols, indicating a strong relationship between the content of (poly)phenols and the antibacterial activity [22,23]. The mechanism the of antimicrobial activity of (poly)phenols has been widely studied, and it has been attributed to the targeting of bacterial cell constituents (cell wall, cell membrane, etc.), interference with bacterial metabolites and ion equilibria, the inhibition of biofilm formation, and interference with nucleic acid synthesis and the regulation of gene expression [23,33].

To the best of our knowledge, this is the first study that tested the antimicrobial efficacy of wine lees against *L. monocytogenes* and *S. cerevisae* in fresh juices. Previously, Alarcón et al. [34] studied the antimicrobial activity of wine lees on deer burger meat and described lees as effective against aerobic psychrotrophic bacteria that survive at refrigeration temperatures. In contrast, the content of β-glucans and mannoproteins from the cell wall of autolyzed yeasts, whose components are part of the wine by-product, could be a carbon source for lactic acid bacteria [35]. In this sense, Ayar et al. [36] studied the use of WL as a prebiotic source for lactic acid bacteria and showed an increase in the growth of probiotics such as *Lactobacillus acidophilus* (ATCC 4357D-5) and *Bifidobacterium animalis* subsp. *lactis* (ATCC 27536) when inoculated in ice cream containing wine lees.

The inhibitory effects of grapes and wine lees have already been assessed against the growth of some food pathogens, such as *L. monocytogenes*, *Staphylococcus aureus*, *Salmonella enterica* subsp. *Enterica* serovar *Typhimurium*, and *Escherichia coli* O157:H7, thereby demonstrating their potential to preserve and prolong the shelf life of food [37].

In agreement with the present results, some authors evaluated the effect of grapes on the inhibition of *L. monocytogenes* and also obtained a good control over even its inhibition [37,38]. Gouvinhas et al. [39] investigated the in vitro antimicrobial activity of grapes and found, as well, inhibitory effects against *L. monocytogenes*. The inhibitory effect of grapes on food pathogens has been attributed to the synthesis of exopolysaccharides, which could be responsible for the inhibition of motility, adhesion, and biofilm formation [37]. In addition, the inhibitory effect of grapes on the growth of this pathogen could also be attributed to their high content of some (poly)phenols with already-demonstrated antimicrobial effects, such as ferulic acid, resveratrol, and gallic and caffeic acid [39]. These phenolic compounds have also been identified in the WL used in our study. Therefore, they could also be responsible for the effects shown by WL in the present study. Moreover, the antimicrobial effect against *L. monocytogenes* of the control peach and grape juice, without WL, may be attributed to the low pH, as suggested by Huang et al. [40], which was also reported for pineapple juice [41].

*S. cerevisiae* is one of the typical yeasts associated with the spoilage of fruit juices [14]. In general, fruit juices, particularly grape products and their resulting beverages, have reported no inhibitory effects against *S. cerevisiae* [42]. Some grape (poly)phenol compounds, such as resveratrol, have shown antimicrobial activity against many microorganisms, including yeasts such as *S. cerevisiae* [43]. Nevertheless, (poly)phenol antimicrobial activity depends on the group and structure of the bioactive and microorganism type, species, and genera [44]. The antimicrobial activity of phenols is based on their ability to disrupt cell membranes and depends on the site and degree of hydroxylation [45]. Due to their cell wall composition, yeasts are more resistant to polyphenol antimicrobial activity than other microorganisms. In addition, not only might yeast structure and resistance to wine lees (poly)phenols contribute to the low antimicrobial efficacy of wine lees in PGJ against this spoilage microorganism reported in the current work, but the carbon-rich compounds in the lees that may be a nutritional source for *S. cerevisiae* might also contribute.

Based on antimicrobial results, the doses of WL, that are more effective *against L. monocytogenes* (1.5 and 2.0%), were selected for the subsequent physicochemical, nutritional, microbiological, and sensory evaluations. The visual appearance of the tested juices is shown in Figure 3.

### 3.2. Effect of the Addition of Wine Lees on the Quality of Peach and Grape Juice

#### 3.2.1. Physicochemical Parameters

For the physicochemical characterization, the pH, total soluble solids (TSS), and titratable acidity (TA) of the three refrigerated juices were monitored for 21 days (Table 1). It could be observed that the addition of WL did not modify the pH of the juice, with a mean of 4.04 ± 0.05. The TA increased moderately, but significantly, with the addition of WL, from 2.96 to 3.85 g malic acid/L for the control and the 2% WL juice, respectively. Significant differences were detected in the TSS when the WL concentration was increased. Thus, the TSS values increased from an average of 13.92 °Brix in the control juice to 15.27 °Brix in the juice with the highest concentration of WL (2%). No significant differences were observed in any of the quality parameters during the storage.

As in many fruits, the pH of peaches is rather acidic because of their content in acidic compounds, mainly malic or tartaric acid [17]. Hence, this parameter was not altered with the addition of the WL, also due to their acidic pH. The titratable acidity (TA) results showed no significant variations over 21 days. The stability of the acidity could suggest a buffered environment that can be supported by the mannoprotein content of the wine lees, resulting from the autolysis of the cell walls of the yeasts that participate in the fermentation processes [6,46,47]. The increase in total soluble solids (TSS) with the addition of WL (1.5% and 2%) can be attributed to the dietary fiber composition of lees, considering the primary element in their composition and also the maltodextrin included in the product formulation. Some authors consider dietary fiber content to be between 22 and 50%, coming from the β-glucans of the yeast cell walls and varying by factors such as grape variety, type of yeast, lees recovery, and winemaking process [6,46]. No evolution of SST was found, either, during the storage time (21 days).

#### 3.2.2. Addition of WL on Total Polyphenols Content (TPC) and Antioxidant Activity (AOX) of Peach and Grape Juice

The evolutions of the TPC and AOX, contents were studied over 21 days of refrigerated storage for two different concentrations of WL (1.5 and 2%).

The addition of the WL to the control juice resulted in a significant increase in the TPC, while the levels were stable throughout the study period (Figure 4).

The incorporation of WL in PGJ increased the total polyphenol content from 11.9 to 36.3 and 46.9 mg GAE/100 mL f.w, for the control, juice with 1.5% of WL, and juice with 2% of lees, respectively. This implied an increase in polyphenol content between 67 and 75% when WL (1.5 and 2%) were incorporated, respectively. Although a slight decrease in phenols was observed during the storage, this was not significant and may be attributed to their oxidative degradation or consumption by the present microorganisms [48]. Contrary to the exhibited results, it has also been reported that phenolic compounds may increase under appropriate storage conditions in fresh juices [49].

The TPC of the juices enriched with WL might be considered similar to that of fruit reported to be rich in antioxidants, such as blueberries, blackberries, and raspberries with values described between 26.7 and 961 mg GAE/100 mL f.w [13]. The increase in polyphenol content, when WL were added, can be justified because one of the main components of these WL, in addition to tartaric acid and autolyzed yeast compounds, are phenolic compounds [37]. The most representative content of phenolic compounds in wine lees corresponds to flavonoids, originally from grapes, which remain adsorbed on the yeast cell wall [17,50]. The content of the total polyphenols of wine lees has been reported to be between 1.9 and 16.3 g/kg depending on the type of grape, fermentation, and processing [51]. The (poly)phenol characterization of the WL by HPLC reported a concentration of 4.5 g/kg. (Poly)phenols present in grapes and grape juices have been shown to exert beneficial effects on health, although not at the level of wine or grape-fermented products. Factors such as organic processing have also been shown to increase the quantity of phenolic compounds in these juices [50].

Regarding the antioxidant activity, both methodologies used for its determination, ferric reducing antioxidant power (FRAP) (Figure 5) and scavenging activity assay (ABTS) (Figure 6), showed the same tendency as the phenols, with an important increase in the antioxidant levels as a function of the increase in the concentration of the WL in the juice. The total antioxidant capacity, determined with ABTS methodology, was enhanced on average from 5.3 mg Eq AA/100 mL f.w to 30.89 mg Eq AA/100 mL f.w and 39.16 mg Eq AA/100 mL f.w, from the control juice to juices with 1.5% of WL and 2% of WL, respectively. This increase in total antioxidant capacity corresponded to an intense increase of 482% and 638%, respectively. In relation to FRAP determination, similar values were observed; the control juice presented an average of 4.07 mg AA Eq/100 mL f.w compared to 14.80 mg AA Eq/100 mL f.w and 25.57 mg AA Eq/100 mL f.w for the juices with 1.5% and 2% of WL, respectively. Such improvement in the antioxidant capacity represented an increase of 263% and 527% for the juice with 1.5% and 2% of WL, respectively. Fruit and vegetable juices contain antioxidants, mostly (poly)phenols, in addition to ascorbic acid, which together contribute to the total antioxidant activity [52]. The role of maltodextrin in these increases was unknown since its effect was not tested independently. Phenols are the major source of antioxidant capacity in peaches [53]. However, fruits that are particularly rich in antioxidants and, therefore, in (poly)phenols, are those red-/purple-/pink-/blue-colored varieties, in which their pigments are typically the major antioxidant source [54]. Anthocyanins are water-soluble pigments within the flavonoids that belong to the (poly) phenol family responsible for the color of grapes and wines [55]. Grapes are recognized for their high nutritional value because their content of (poly)phenols, and flavonoids contribute the most to their antioxidant activity [56]. It has even been reported that flavonoids may function as antioxidants, apparently through metal chelation [12]. Studies even reflect a possible increase in antioxidant activity promoted by the content of extractable and non-extractable total phenols, which may correlate positively with the total polyphenol content of grapes [47].

In addition to the bioactive compounds typically present in fruits such as peaches and grapes, which could explain the antioxidant activity of the samples, studies on fermented beverages such as wine, beer, and juices confirm a significant additional antioxidant power for these products, mainly attributed to the fermentation process [12]. The previously reported increment in antioxidant capacity during fermentation places wine lees as a valuable source of bioactive compounds with antioxidant capacity due to the fact that they are the by-product of the fermentation of grapes [57]. This might explain the major increment of polyphenol content and antioxidant activity observed when a small amount of WL was added to the juice.

To further evaluate the scavenging effects of the antioxidants present in the juices, the maximum mean inhibitory concentration (IC_50_) was determined (Figure 7). An IC_50_ value of 326.02 mg/mL was found for the control juice, while the juices with WL (1.5% and 2%), resulted in significantly lower values: 69.84 and 60.89 mg/mL, respectively. These results suggest that a higher amount of control juice may be necessary to scavenge 50% of the free radicals. Lower IC_50_ values indicate higher radical scavenging activity and, therefore, higher AOX, which was in line with the results observed for polyphenol content and antioxidant capacity. Lower values for IC_50_ activity have already been reported for grapes and wine subproducts than for peaches [57,58], which may explain the significantly lower values found in juices with WL. Furthermore, De Santis et al. [59] described a reduction of approximately 50% in the IC_50_ when comparing control pasta with pasta enriched with a by-product of raspberry juice, which may support the application of WL in juices to increase antioxidant activity and, therefore, functional properties.

The degree of fruit processing is a factor affecting polyphenol content, and, thus, antioxidant capacity, which can result in the loss of or increase in some bioactive compounds, also impacting their bioaccessibility and bioavailability [60,61]. Processes such as clarification and stabilization can influence the decrease in flavonoids, while the oxidative degradation of (poly)phenols can occur in pulp processing, thanks to the release and action of the enzyme (poly)phenol oxidase [62]. The degradation of antioxidant substances and other bioactive compounds that may result from the thermal and mechanical treatments of the juices, such as pasteurization or clarification, influenced the decision not to subject the juice to further processing.

Storage temperature is a determining variable for the microbiological stability of fruit juices and their antioxidant composition. Some authors [49] reported an increase in phenolic and other bioactive compounds when fruit juices were fresh and stored under appropriate conditions.

Moreover, ecological processes such as the maceration or pressing of the fruit can result in the diffusion of antioxidants in the juice. The above references may support the advantages of minimally processed and ecological juices rich in bioactive compounds. Taking into account the antioxidant activity that many of the (poly)phenols, and in particular wine lees (poly)phenols, supply, the study of correlations between TPC and AOX (FRAP and ABTS) was considered to be of interest (Table 2).

In agreement with Romero-Díez et al. [57], who described a positive correlation between TPC and AOX (r^2^ > 0.9, *p* value < 0.05) in WL, the present study established a significant positive correlation (*p* < 0.01) between TPC and AOX with a reducing power assay (FRAP) (r^2^ = 0.9572) and scavenging activity assay (ABTS) (r^2^ = 0.9805) (Table 2). The relationship between the antioxidant activity and phenolic content depends on several factors, such as the chemical structure of the individual component, the synergistic interaction among them, and the specific conditions applied in the different assays [39]. As it has been mentioned and verified with the high positive correlation obtained between TPC and AOX, phenolic compounds are mostly responsible for the total antioxidant activity in fruit and fruit juices. The scavenging of ABTS radical by phenolic compounds depends mainly on the number of hydroxyl groups present in the molecules and their electron-donating ability. Thus, the closer correlation between AOX scavenging activity assay (ABTS) and TPC (r^2^ = 0.9805) that was found could be explained. Various authors have already reported the correlation between the total polyphenol content and antioxidant activity in fruit juices [9,63,64]. This could explain the stability of antioxidant power and polyphenol content values, through the storage time, and the low positive correlation between TPC and AOX (FRAP and ABTS).

### 3.3. Microbiological Quality

The total mesophilic aerobic microorganisms, molds, and yeasts were determined as microbiological quality indicators of the juice and monitored over its shelf life. The initial population of aerobic microorganisms in the juice was approximately 10^4^ CFU/mL (Figure 8), while the addition of WL did not result in a significant inhibitory effect. Opposite to the behavior of *L. monocytogenes* in the presence of WL, the initial population of mesophiles in the juice with lees decreased on day 4, showing significant differences with the control, but afterwards, it increased and, after 7 days, no significant differences between treatments were observed. Between days 7 and 11, the mesophilic population increased in the order of 1–3 log units and almost in the same proportion for days 14 and 21 of refrigerated storage, with an overall average of 2.9 log units (log CFU/mL) during the monitoring period. The final population was between 6.3 and 7.3 log CFU/mL.

Concerning the mold and yeast counts, no molds were observed; therefore, the results represented correspond only to yeasts (Figure 9). In general, the yeast population was remarkably similar regardless of whether the juice contained WL or not and increased throughout the storage. Similar to the mesophilic counts, the yeast count grew at a slower rate during the first 7 days of the trial and significantly increased in proliferation by day 11 with 1 to 2 log units and at the same rate until day 21. It should be noted that the initial growth of mesophiles was twice that of the yeasts, with 2 and 4 log CFU/mL, respectively. However, the yeasts managed to increase their population over the 21 days of monitoring, with almost 5 log units compared to the bacteria which increased their population by only 3 log units.

The total aerobic mesophilic and yeast counts grew in the peach and grape juice. It has been reported that yeasts can use glucose from the fruit extracts and the dietary fiber of the WL as a source of carbon from glucose, mannose, and rhamnose [6], as verified in our results, in which the yeast population increased by almost 5 log units during the refrigerated storage time, regardless of the addition of WL. Similarly, Leneveu-Jenvrin et al. [41] have already demonstrated that psychrotrophic bacteria, yeast, and molds increased during the refrigerated storage of unpasteurized pineapple juices, as well as in other apple beverages [65]. Contrary to what was observed with *L. monocytogenes*, a Gram-positive bacterium, the addition of lees did not affect the growth of the indigenous microorganisms found in the juice, in special yeasts, since no significant differences were observed between the sample without lees and the sample with lees. It is true that the total aerobic mesophilic counts were significantly lower in the juice with WL after 4 days of storage, but the population was the same after 7 days. This could be attributed to the bacteriostatic effect of polyphenols (poly)phenols, present in peaches, grapes, and lees. It has been found that Gram-negative bacteria, molds, and yeasts are more resistant to bactericidal polyphenols than Gram-positive bacteria [66]. Finally, it has already been described how several nutrients present in juices may support the growth of acid-tolerant bacteria, yeasts, and molds, which would be proven by the higher values of these microorganisms found throughout life in the current research [67].

Although there is not a formal legal limiting mesophilic aerobic, molds, or yeast count in juices according to the European Commission Regulation (No. 2073/2005 and No. 1441/2007) [68] and Spanish microbiological criteria (No. 3484/2000) [69], prepared meals without heat treatment and heat-treated prepared meals containing non-heat-treated ingredients, like in this case, the recommended mesophilic aerobic threshold value would be 10^5^ CFU/mL for. Therefore, juices after day eleven could be considered highly contaminated. Consequently, the shelf life of fresh peach and grape juice with or without wine lees would be around 10 days. Consequently, pasteurization is recommended if this formula is to be commercialized.

### 3.4. Sensorial Analysis

Although their nutritional potential (TPC and AOX) and microbiological effects can be challenging due to the sensory impact on the final food products to which they are added, WL are used as a functional ingredient. In the sensory evaluation of the three samples tested, juices with lees scored lower (5.5 points) than the control juice (7.2 points) in the overall evaluation of their organoleptic characteristics. However, 69% of consumers gave it a score of more than 5 out of a total of 9 points, which may indicate that this product, although it is a new formulation, may have a target consumer group. Regarding taste, all three PGJs also obtained satisfactory scores (Figure 10). However, while the PGJ control was accepted by 92% of tasters (scores equal to or higher than 3), only 57% and 50% of consumers rated the PGJs positively with 1.5 and 2% WL, obtaining a mean score of 2.9 and 2.8, respectively. Some panelists who did not like the juice appreciated some notes related to wine or fermented products. Previously, WL have been incorporated into food matrices such as sourdough and hamburger meat [6,33]. The burgers managed to retain the meat’s own color and presented wine and bread notes, considered pleasant at low intensity due to the increase in benzene compounds, esters, and acids present in the WL that impact the volatile components of the product and, consequently, characteristics such as odor and flavor [33]. In the case of bread, the addition of WL had very good consumer acceptance [6]. Although the addition of WL resulted in a favorability decline from some of the panelists owing to the fermented taste/aroma, the functional properties of juice fortified with WL could increase consumers’ inclination to purchase. Previous studies have demonstrated the preventive action of the bioactive compounds present in wine lees against the oxidation of macromolecules and consequently, oxidative stress that can lead to cardiovascular pathologies, diabetes, cancer, and other diseases associated with aging. The lipid-lowering and antihypertensive activities of wine lees reported in animal models could encourage, as well, the consumption of juices or any food product fortified with them [18].

Another appreciation considered by the panelists was the attribute “texture” for which observations related to thickness or density were made. These observations could be explained by the fiber content of the WL and the fact that it is an unclarified, smoothie-type fresh juice. The dietary fiber content in wine lees contributes to some of their techno-functional properties, such as water retention capacity (WRC) and fat retention capacity (FAC) [46].

Although the two juices with WL were not widely accepted, attributed to their unpleasant texture, or fermented notes reported by consumers, their physicochemical composition and functionality place WL as a by-product with potential in the food industry as a technological adjuvant and functional ingredient. Future studies are needed to determine whether these evaluated effects of WL can be extrapolated to humans, to consider them as a functional ingredient at the food level, or in the development of new nutraceuticals or pharmaceuticals.

## 4. Conclusions

The results of this study showed that the inclusion of WL in fresh peach and grape juice is an opportunity for the valorization of this by-product of the winemaking process as a functional ingredient; meanwhile, the juice could be a great opportunity to extend the shelf life of peach fruit and add nutritional and antimicrobial value to it. Under refrigerated storage conditions, the incorporation of the WL improved the nutritional profile (polyphenols and total antioxidants) of the product and maintained stable physicochemical variables, such as pH and titratable acidity. Although the addition of lees did not have an evident inhibitory effect on the population of total mesophilic microorganisms, this effect was observed against *L. monocytogenes*, which is one of the most concerning foodborne pathogens in refrigerated ready-to-eat products. Despite its significant nutritional value, juice containing WL was not widely accepted. Likewise, it is necessary to highlight that it seems to be a market niche for this product composed of health-conscious consumers. Based on these findings, the bioactive compounds in WL incorporated into juice could be employed as a potential source of natural antioxidants and antimicrobials. The utilization of the fruit industry by-products as functional products may be an effective tool for promoting health and preventing foodborne diseases. This bioactive-enriched alternative product opens an interesting field in the industry since it applies strategies for the valorization of juices, already on the market, and considers the potential of by-products, such as WL, for the elaboration of new functional food goods.

Further studies, including the characterization of polyphenols and other bioactive compounds demonstrating antimicrobial activity and functional properties of lees and their potential as food ingredients should be conducted.

## Figures and Tables

**Figure 1 foods-13-01095-f001:**
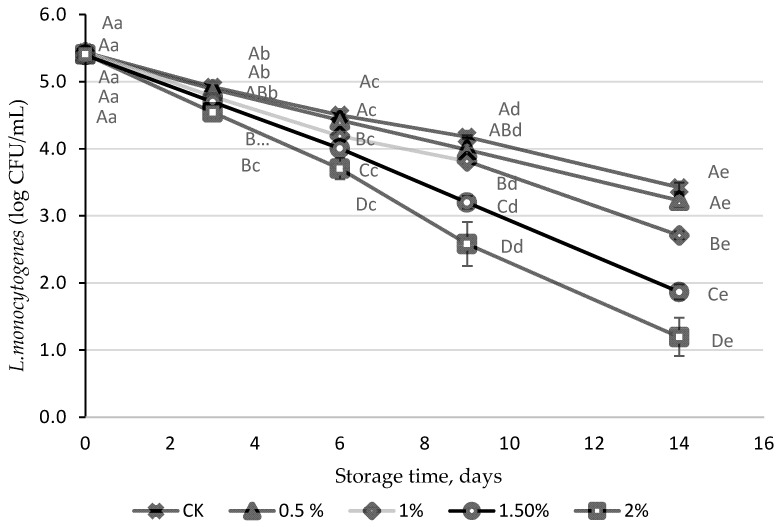
*L. monocytogenes* population (log CFU/mL) in a peach and grape juice (50:50, *v*:*v* CK), and in the same juice but enriched with 0.5, 1.0, 1.5, and 2% of wine lees (WL) stored at 5 °C. For each day of storage, different capital letters indicate significant differences among different WL. For each juice, different lowercase letters indicate significant differences between the different days of storage according to an ANOVA test (*p* < 0.05).

**Figure 2 foods-13-01095-f002:**
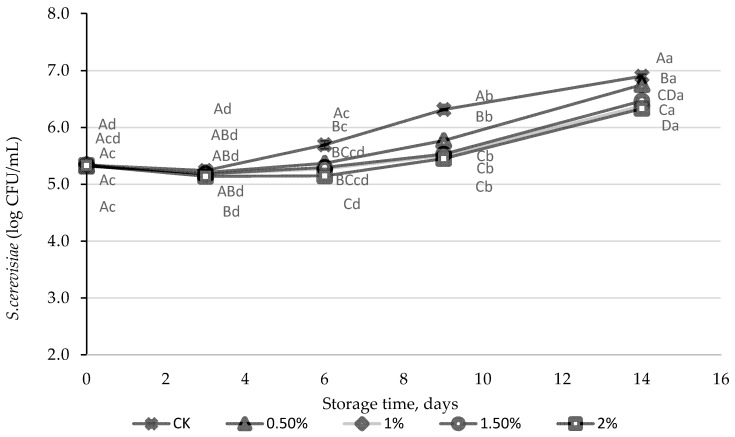
*S. cerevisae* population (log CFU/mL) in peach and grape juice (50:50, *v*:*v*, CK), and in the same juice enriched with 0.5, 1.0, 1.5, and 2% of wine lees (WL) stored at 5 °C. For each day of storage, different capital letters indicate significant differences between different WL concentrations. For each juice, different lowercase letters indicate significant differences between the different storage days according to an ANOVA test (*p* < 0.05).

**Figure 3 foods-13-01095-f003:**
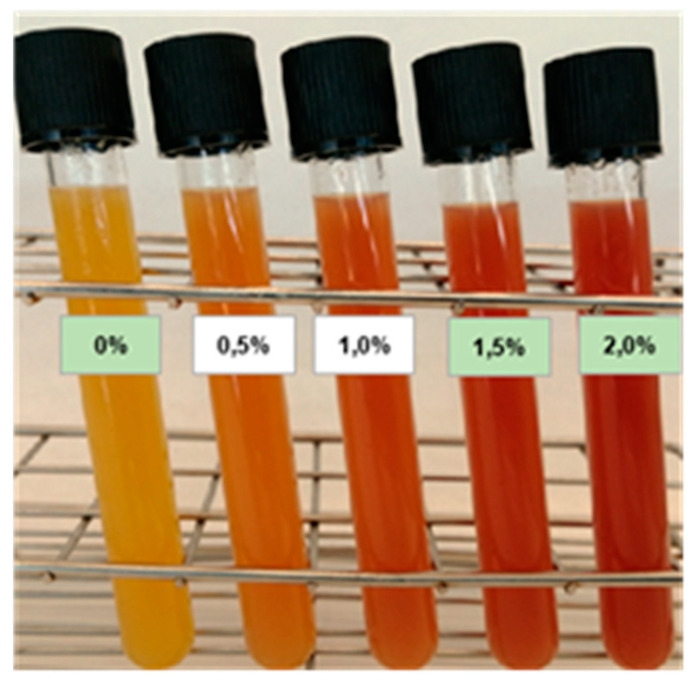
The visual aspect of the juices: the control juice (WL 0%) and the different wine lees tested doses (WL: 0.5, 1.0, 1.5, and 2.0%).

**Figure 4 foods-13-01095-f004:**
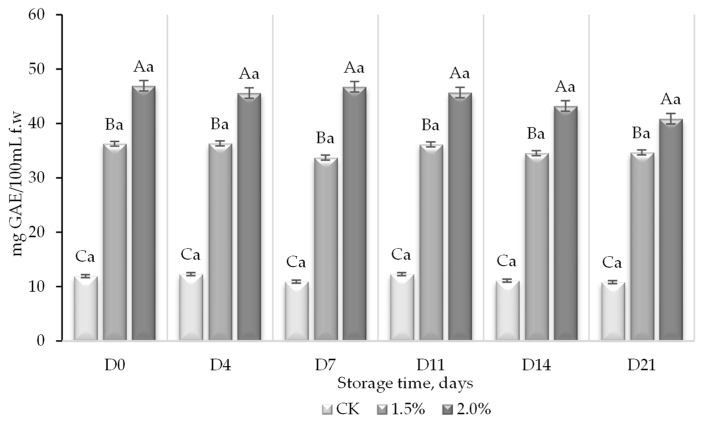
Total polyphenol content (TPC) (mg equivalents of gallic acid (mg GAE)/100 mL f.w) in a peach and grape juice (50:50, *v*:*v*). CK (0% of WL) and the same juice enriched with 1.5 and 2% WL and its evolution during storage at 5 °C. For each day of storage, different capital letters indicate significant differences between different WL concentrations. For each juice, different lowercase letters indicate significant differences between the different storage days according to an ANOVA test (*p* < 0.05).

**Figure 5 foods-13-01095-f005:**
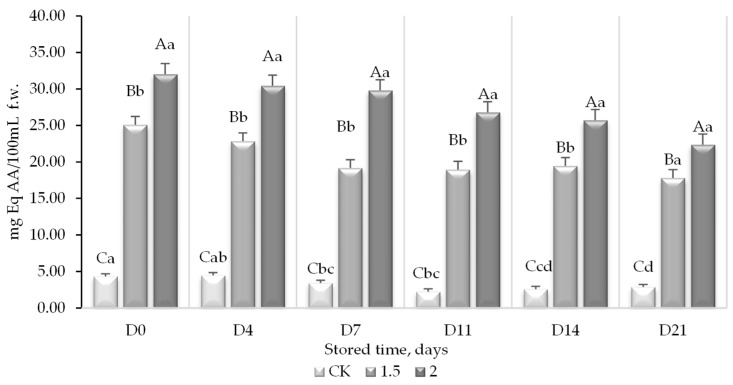
Total antioxidant capacity (AOX) measured by the FRAP method (mg equivalent ascorbic acid/(EqAA)/100 mLf.w) in peach and grape juice (50:50, *v*:*v*; CK) and in the same juice enriched with 1.5 and 2% of wine lees (WL) and its evolution during storage at 5 °C. For each day of storage, different capital letters indicate significant differences between WL concentrations. For each juice, different lowercase letters indicate significant differences between storage days according to an ANOVA test (*p* < 0.05).

**Figure 6 foods-13-01095-f006:**
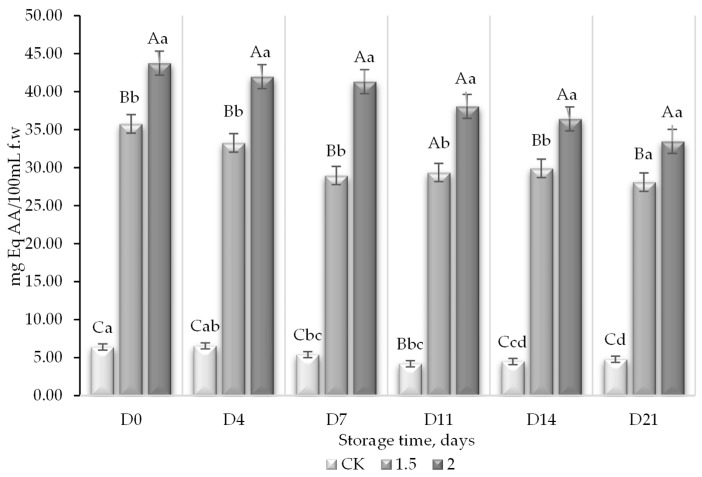
Total antioxidant capacity (AOX) measured by ABTS method (mg equivalent ascorbic acid (EqAA)/100 mL f.w) in peach and grape juice (50:50, *v*:*v*; CK) and the same juice enriched with 1.5 and 2% wine lees (WL) and its evolution during storage at 5 °C. For each day of storage, different capital letters indicate significant differences between WL concentrations. For each juice, different lowercase letters indicate significant differences between the different days of storage according to an ANOVA test (*p* < 0.05). As observed for the TPC, AOX significantly increased with the addition of lees to the juice and levels did not decrease during 21 days of storage at 5 °C.

**Figure 7 foods-13-01095-f007:**
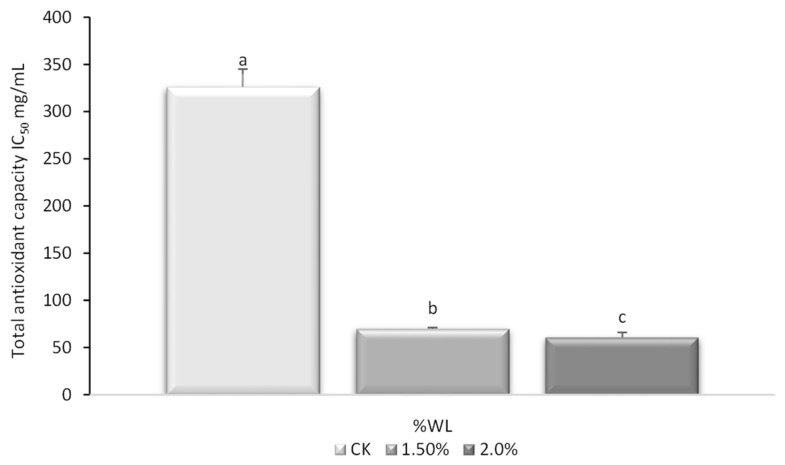
Total antioxidant capacity (AOX) measured by ABTS method and expressed as IC_50_ (mg/mL extract) in peach and grape juice (50:50, v/v; CK) and the same juice enriched with 1.5 and 2% of wine lees (WL) and its evolution during its conservation at 5 °C. Different lowercase letters indicate significant differences between WL concentrations according to an ANOVA test (*p* < 0.05).

**Figure 8 foods-13-01095-f008:**
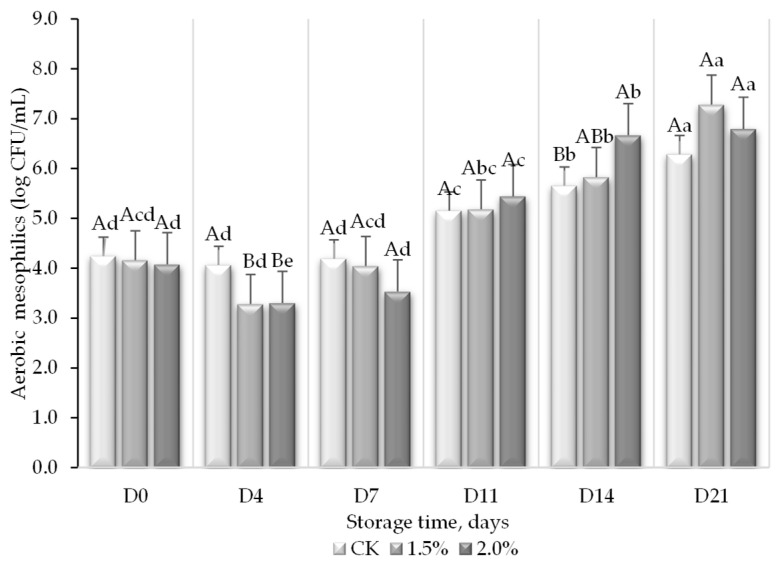
Total aerobic mesophilic counts (log CFU/mL) in peach and grape juice (50:50, *v*:*v*; CK) and the same juice enriched with 1.5 and 2% wine lees (WL) and their evolution during storage at 5 °C. For each day of storage, different capital letters indicate significant differences between WL concentrations. For each juice, different lowercase letters indicate significant differences between the different days of storage according to an ANOVA test (*p* < 0.05).

**Figure 9 foods-13-01095-f009:**
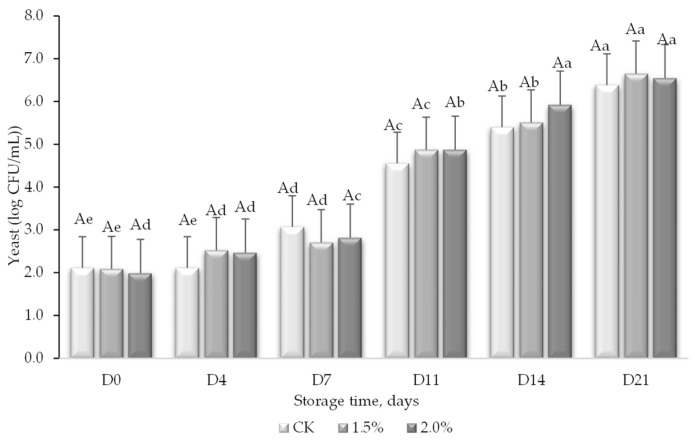
Yeast counts (log CFU/mL) in peach and grape juice (50:50, *v*:*v*; CK) and the same juice enriched with 1.5 and 2% wine lees (WL) and their evolution during storage at 5 °C. For each day of storage, different capital letters indicate significant differences between WL concentrations. For each juice, different lowercase letters indicate significant differences between the different days of storage according to an ANOVA test (*p* < 0.05).

**Figure 10 foods-13-01095-f010:**
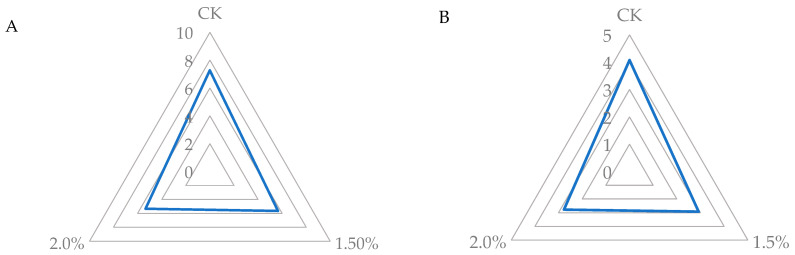
Sensory evaluation of the overall “organoleptic characteristics” (**A**) and the attribute of “flavor” (**B**) in a peach and grape juice (50:50, *v*:*v*; CK) and the same juice enriched with 1.5 and 2% of WL. “Organoleptic characteristics” were evaluated on a scale of 1 to 9 and the attribute “flavor” on a scale of 1 to 5.

**Table 1 foods-13-01095-t001:** Changes in SSS (°Brix) and TA (g malic acid/L of juice) of peach juice.

	Sample/Days	0	4	7	11	14	21
SST	CK	13.93 ± 0.06Ca *	13.97 ± 0.06Ca	13.93 ± 0.06Ca	13.90 ± 0.00Ca	13.87 ± 0.02Ca	13.87 ± 0.06Ca
	1.5	14.70 ± 0.08Bb	14.77 ± 0.06Bab	14.70 ± 0.08Bb	14.83 ± 0.06Ba	14.77 ± 0.01Bab	14.73 ± 0.06Bab
	2	15.27 ± 0.06Ab	15.30 ± 0.00Ab	15.27 ± 0.06Ab	15.27 ± 0.06Ab	15.47 ± 0.01Aa	15.20 ± 0.00Aa
TA	CK	2.96 ± 0.05Cab	2.85 ± 0.01Cb	2.93 ± 0.198Bab	2.93 ± 0.19Cab	3.11 ± 0.13Cab	3.51 ± 0.48Aa
	1.5	3.56 ± 0.04Ba	3.61 ± 0.48Ba	3.56 ± 0.14Aa	3.56 ± 0.14Ba	3.57 ± 0.14Ba	3.36 ± 0.3Aa
	2	3.73 ± 0.09Ab	3.82 ± 0.02Ab	3.84 ± 0.08Aab	3.84 ± 0.08Ab	3.86 ± 0.05Aab	4.02 ± 0.03Aa

(*) For the same parameter, different capital letters denote significant differences between treatments, and different lowercase letters denote significant differences between days (*p* < 0.05). Values correspond to the mean of 3 replicates ± standard deviation. CK: PGJ without WL; 1.5% and 2% concentration (%, *w*/*v*) of WL product incorporated in the PGJ.

**Table 2 foods-13-01095-t002:** Pearson correlation coefficients r^2^ of TPC and antioxidant activity (FRAP and (ABTS) in PGJ (50:50, *v*:*v*; CK) and the same juice enriched with 1.5 and 2% of wine lees during 21 days of storage at 5 °C.

	TPC	ABT	FRAP
**TPC ***	1.000	0.981	0.958
**ABTS**	0.981	1.000	0.916
**FRAP**	0.958	0.916	1.000

*TPC refers to total polyphenol content; ABTS to the radical scavenging assays using 2,2′-azino-bis (3-ethylbenzothiazoline-6-sulfonic acid) and FRAP to Ferric Reducing Antioxidant Power Assay

## Data Availability

The original contributions presented in the study are included in the article, further inquiries can be directed to the corresponding author.

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
