# Peer review of "Valorization of Peach Fruit and Wine Lees through the Production of a Functional Peach and Grape Juice"

_foods, 2024, doi:10.3390/foods13071095_

Round 1
Reviewer 1 Report
Comments and Suggestions for Authors
Please correct as attached

Need minor correction
Author Response
Only grammatical issues were indicated by the reviewer and all suggestions were addressed.

Reviewer 2 Report
Comments and Suggestions for Authors
The effects of wine lees on the composition and antioxidant activity of peach juice were investigated in the present MS. The topic is interesting. Some issues should be addressed before further processing.
1. Table 1. Physicochemical parameters of the juices: pH, TSS (°Brix), and TA. The data of pH were mot found in Table 1. In this case, the Table title should be changed as “Changes in TSS (°Brix) and TA (g malic acid/L of juice) of peach juice.
2. Table 2: Statistically significant levels should be included in the data of correlation coefficients.
3. (A) and (B) should be labeled in the Figure 11.
4. More explanations should be given to the abnormal experimental data such as those in Figures 9-10. In the early incubation stage such as at 7th day, the addition of lees inhibited the growth of microorganism, but it promoted the growth of microorganism in the late stage. Normally, polyphenols have inhibiting effects on growth of microorganism. The readers might not understand these phenomena and the authors should help the readers to understand them.
5. The MS is not organized well, such as Section 3.2. Total Polyphenols Content (TPC), Total Antioxidant Capacity (AOX) and Vitamin C. determinations. Vc is also a parameter of chemical. This section should be divided into two sections. i.e., Addition of lees on chemical composition of peach juice, and addition of less on antioxidant activity of peach juice.
6. Conclusion: the descriptions which were not related to the conclusions should not be included in this section, such as shoes in Lines 728-733.
Author Response
Reviewer 2:
The effects of wine lees on the composition and antioxidant activity of peach juice were investigated in the present MS. The topic is interesting. Some issues should be addressed before further processing.
- Table 1. Physicochemical parameters of the juices: pH, TSS (°Brix), and TA. The data of pH were not found in Table 1. In this case, the Table title should be changed as “Changes in TSS (°Brix) and TA (g malic acid/L of juice) of peach juice.
As mentioned in the text: “the addition of WL did not modify the pH of juice, with a mean of 4.04 ± 0.05 (data not shown)”, for that reason it was decided to delete “pH” from the title of the table.
- Table 2: Statistically significant levels should be included in the data of correlation coefficients.
Statistically significant level for every correlation of p < 0.05 was included in the table footer.
- (A) and (B) should be labeled in the Figure 11.
(A) and (B) were added in the figure.
- More explanations should be given to the abnormal experimental data such as those in Figures 9-10. In the early incubation stage such as at 7th day, the addition of lees inhibited the growth of microorganism, but it promoted the growth of microorganism in the late stage. Normally, polyphenols have inhibiting effects on growth of microorganism. The readers might not understand these phenomena and the authors should help the readers to understand them.
In the early incubation stage such as at 7th day, the addition of lees inhibited the growth of microorganism, but it promoted the growth of microorganism in the late stage. Normally, polyphenols have inhibiting effects on growth of microorganism.
There was an inhibition of the growth of total aerobic mesophilic counts – not the yeasts – at D4, as statistically differences were found. This could be due to the bacteriostatic effect of polyphenols in relation to the cause a longer lag phase and reducing the growth rate. However, these differences were not observed after 7, 11, and 21 days. Even statistically differences were observed in the D14, we did not consider as important since they do not follow the pattern observed along the storage.
We have changed the text accordingly:
Contrary to what was observed with L. monocytogenes, a gram-positive bacteria, the addition of lees did not affect the growth of the indigenous microorganisms found in the juice, in special yeasts, since no significant differences were observed between the sample without lees and the sample with lees. It is true that total aerobic mesophilic counts were significantly lower in the juice with WL after 4 days of storage, but population was the same after 7 days. This could be attributed to the bacteriostatic effect of polyphenols (poly)phenols, present in peach, grape and lees. It has been found that gram-negative bacteria, moulds and yeasts are more resistant to bactericidal polyphenols than gram-positive bacteria (Fattouch et al., 2007).
- "The MS is not organized well, such as Section 3.2. Total Polyphenols Content (TPC), Total Antioxidant Capacity (AOX) and Vitamin C. determinations. Vc is also a parameter of chemical. This section should be divided into two sections. i.e., Addition of lees on chemical composition of peach juice, and addition of less on antioxidant activity of peach juice.
This consideration has been taken into account. However, this section was designed and written accordingly so that the reader would understand the possible impact of polyphenol content on antioxidant capacity. Furthermore, considering that both polyphenolic compounds and vitamin C may have some impact on antioxidant capacity, a final correlation was performed, emphasizing the positive impact of the addition of WL to PGJ. The specifications of "chemical composition and antioxidant activity" were included in the section title for further clarity.
- Conclusion: the descriptions which were not related to the conclusions should not be included in this section, such as shoes in Lines 728-733.
The conclusions section has been revised and the suggested consideration has been addressed. Lines 728 -733 have been removed.
Reviewer 3 Report
Comments and Suggestions for Authors
Line 18. % of what w/w, w/v, v/v? Please clarify.
Lines 39-42. Please rephrase.
Line 121. Please correct me if I am wrong. The WL product was actual 47.1% WL and 52.9% maltodextrin. If it is correct, I think that this is a problem. Large amount of maltodextrin.
Figures. “(log(CFU/mL)” Please delete the extra parenthesis. (logCFU/mL)
Table 1. Where is pH? SST?
Lines 554-555. This is not an appropriate explanation of higher vit C content. The reference provided is about table grapes. WL are selected after the completion of fermentation. In my opinion the vitamin C content there is too low. Especially after spray drying of the WL product. Please be careful. If the analysis is not correct delete all the results of vitamin C.
Figure 11. I do not think that this is the ideal presentation. Please delete and provide values either in the text or in a table.
Materials and methods. I cannot find the production details of juice. Please add.
Please explain the necessity to add grape juice in the peach juice.
Lines 66-77. I do not think that it is so important the content of vit C. It is very easy to add vit C in the final juice. Please also discuss the results of figure 8. 14 μg/100mL are important? I think it is very low to even discuss it.
In general the stability of the final juice is very low. After one week problems occur.
In my opinion it would be better to have a pasteurized product and evaluate all these parameters. The high numbers of bacteria and moulds have as a result fermentations to occur.
The main concern is the addition of high amount of maltodextrin. More than 50% of WL were actually maltodextrin. Is there any effect on that?
I have also concern regarding the title of the manuscript. I think that the addition of grape juice should be incorporated. In the present form the readers understand the there would be a peach juice only with WL.
Comments on the Quality of English LanguageMinor
Author Response
- Line 18. % of what w/w, w/v, v/v? Please clarify.
This suggestion was added to the test.
- Lines 39-42. Please rephrase.
This sentence was rephased as following:
“Functional foods not only stand out for their nutritional value, but also for playing an important role in promoting optimal health, even preventing some non-communicable diseases such as cancer or cardiovascular pathologies”.
- Line 121. Please correct me if I am wrong. The WL product was actual 47.1% WL and 52.9% maltodextrin. If it is correct, I think that this is a problem. Large amount of maltodextrin.
Yes, the concentration of WL on the product is 47.1 %. Maltodextrin was used as a carrier, needed for the spray-drying process. Maltodextrin has been widely used for the encapsulation of bioactive compounds to increase their stability and to preserve their functional potential, due to the low cost, high solubility, neutral flavuor, and encapsulation efficiency. Maltodextrins have been used in fruit juice-based beverages to improve flavour and provide mouth feel (Featherstone, 2015, https://doi.org/10.1016/B978-0-85709-678-4.00008-7). Therefore, from our point of view the maltodextrin used in the formulation of the WL should not be a problem, but on the contrary, since it can help to maintain bioactivity.
- “(log(CFU/mL)” Please delete the extra parenthesis. (logCFU/mL)
This mistake has been corrected in this line and through all the manuscript.
- Table 1. Where is pH? SST?
This mistake has been addressed and pH was deleted from the table title. As it was indicated in the manuscript and since pH values were stable throughout storage, we decided not to show it.
- Lines 554-555. This is not an appropriate explanation of higher vit C content. The reference provided is about table grapes. WL are selected after the completion of fermentation. In my opinion the vitamin C content there is too low. Especially after spray drying of the WL product. Please be careful. If the analysis is not correct delete all the results of vitamin C.
This consideration was taken into account and the reference was changed.
The significant higher content of vitamin C in juices with lees (1.5, 2%) are probably explained by the content of vitamin C of wine by-product [60].
The reference was changed. As indicated in the reference, grape skins also contain vitamin C, but in lower quantities than grapes do.
- Charalampia, A.Ε. Koutelidakis, A. Koutelidakis, Value Added Alternatives of Winemaking Process Residues: A Health Based Oriented Perspective. BAOJ Biotech, 2016, 2, 016.
- Figure 11. I do not think that this is the ideal presentation. Please delete and provide values either in the text or in a table.
This kind of presentation is usually used for acceptability sensorial analysis. We left it in the manuscript, but we added some more text for clarification.
“The use of WL as a functional ingredient, although their nutritional potential (TPC, AOX, and vitamin C) and microbiological effects can be result challenging due to the sensory impact on the final food products to which they are added. In the sensory evaluation (Figure 11) of the three samples tested, juices with lees scored lower (5.5 points) than the control juice (7.2 points) in the overall evaluation of their organoleptic characteristics. However, 69 % of consumers gave it a score of more than 5 out of a total of 9 points, which may indicate that this product, although it is a new formulation, may have a target consumer group. Regarding taste, all three PGJs also obtained satisfactory scores. However, while the PGJ control was accepted by 92% of tasters (scores equal to or higher than 3), only 57% and 50% of consumers rated the PGJs positively with 1.5 and 2% WL, respectively”.
- Materials and methods. I cannot find the production details of juice. Please add.
This section has been revised and several changes have been made in order to ensure legibility and clarity.
“Peach juice was obtained from yellow flesh peach (Prunus persica / L./ Batch) cv. Royal Summer, in the pilot plant at IRTA's Fruitcentre facilities, using an automatic extractor (Robot Coupe C80A, Mataró, Barcelona, España). This juice contained neither added sugars nor it was subjected to subsequent clarification or pasteurization treatments. The juice was kept frozen (-20 °C) until further use.
Grape juice (ECO-Call Valls, Vilanova de Bellpuig, Lleida, Spain), with no added sugars, was elaborated directly from organically grown grapes of the native ‘Macabeu’ and ‘Parellada’ varieties.
The defrost peach juice and commercial grape juice was mixed 50:50 with to obtain the peach and grape juice (PGJ)”.
A new section ("formulation of functional juices") has also been added for greater clarity.
- Please explain the necessity to add grape juice in the peach juice.
The peach juice obtained with our extractor, was very thick, dense, more similar to a puree texture. For that reason, most of the commercial peach juices in the Spanish market include other juices in their formulation (eg. Grape or apple), are clarified or are processed as nectars (including water and sugar). In our case we decided to mix with grape juice
- Lines 66-77. I do not think that it is so important the content of vit C. It is very easy to add vit C in the final juice. Please also discuss the results of figure 8. 14 μg/100mL are important? I think it is very low to even discuss it.
There was a calculation error. Now the results are expressed in mg TAA/100mL, since as we have seen, most papers indicate vitamin C content in these units. Aguilar et al., 2018 described a similar vitamin C content (0.2-0.9 mg/mL) in juices of 3 peach varieties obtained locally, in the region of Lleida (Catalonia, Spain), where our feedstock also comes from.
We know that the content of Vitamin C in peach juice is not important. However, we wanted to see if the addition of WL improved it, what we demonstrated.
- Aguilar, A. Garvín, A. Ibarz, Effect of UV–Vis processing on enzymatic activity and the physicochemical properties of peach juices from different varieties, Innovative Food Science and Emerging Technologies 48 (2018). https://doi.org/10.1016/j.ifset.2018.05.005.
- In general the stability of the final juice is very low. After one week problems occur.
In my opinion it would be better to have a pasteurized product and evaluate all these parameters. The high numbers of bacteria and moulds have as a result fermentations to occur.
The consideration is right. It is essential to take into account that an unpasteurized juice was employed for the formulation. It is indicated in the manuscript that according to legislation this beverage presents a limited self-life:
“Therefore, juices after the day eleven could be considered highly contaminated. Consequently, the self-life of fresh peach and grape juice with or without wine lees would be around 10 days”.
Therefore, if the product is sold as fresh, it will have a short shelf-life (similar to a fresh-cut product). Instead, a pasteurization step should be carried out in order to obtain a stable final product and commercialize it. Therefore, this line was added to the text as following:
“Consequently, pasteurization is recommended if this formula is to be commercialized.”.
- The main concern is the addition of high amount of maltodextrin. More than 50% of WL were actually maltodextrin. Is there any effect on that?
Maltodextrin was used as a carrier, being essential to obtain a stable ingredient (see comments to Reviewer 2 about the same question). The direct effect of maltodextrin addition in the formulation is correlated with ºBrix, as indicated by the physicochemical results. In addition, the potential impact of this maltodextrin may be associated with its role as a prebiotic.
- I have also concern regarding the title of the manuscript. I think that the addition of grape juice should be incorporated. In the present form the readers understand the there would be a peach juice only with WL.
According to this consideration, the new ssuggested tittle could be “Valorization of Peach Fruit and Wine Lees Through the Production of a Functional Peach and Grape Juice”.
Round 2
Reviewer 3 Report
Comments and Suggestions for Authors
The presence of maltodextrin is not a problem as an additive but for the concept of the study that valorize wine lees. The addition of 50% maltrodextrin and 50% of wine lees should be clearly stated in the text. In addition, the discussion of the results should also take this into consideration. The differences should be attributed also in the addition of maltrodextrin.
Table 1. TSS in title and SST in table. Please correct.
The content of vit C I think that does not provide anything important in the manuscript. The content is relevant too low. If such a juice produced in commercial level the addition of vitamin C is necessary. Also the difference by the addition of wine lees is too low. Please delete.
“The significant higher content of vitamin C in juices with lees (1.5, 2%) are probably explained by the content of vitamin C of wine by-product [60]”. This reference [60] does not provide anything regarding the vit. C content of wine lees. Please do not confuse all the winemaking by products. Grape pomace flour, grape skin are completely different with wine lees.
Comments on the Quality of English LanguagePlease double check the manuscript for minor corrections.
